# Optimization of Experimental Variables Influencing Apoptosome Biosensor in HEK293T Cells

**DOI:** 10.3390/s20061782

**Published:** 2020-03-23

**Authors:** Azarakhsh Oladzad, Maryam Nikkhah, Saman Hosseinkhani

**Affiliations:** 1Department of Biochemistry, Faculty of Biological Sciences, Tarbiat Modares University, Tehran P.O. Box 14115-175, Iran; azarakhsh.oladzad@modares.ac.ir; 2Department of Nanobiotechnology, Faculty of Biological Sciences, Tarbiat Modares University, Tehran P.O. Box 14115-175, Iran; m_nikkhah@modares.ac.ir

**Keywords:** apoptosis, apoptosome, luciferase complementary assay

## Abstract

The apoptotic protease-activating factor 1 (Apaf-1) split luciferase biosensor has been used as a biological tool for the detection of early stage of apoptosis. The effect of doxorubicin in a cell-based assay and the addition of cytochrome *c* and ATP in a cell-free system have been used to test the functionality of the reporter for the detection of apoptosome formation. Here, our data established a drug- and cytochrome *c*/ATP-independent way of apoptosis induction relying on the expression of the biosensor itself to induce formation of apoptosome. Overexpression of Apaf-1 constructs led to increased split luciferase activity and caspase-3 activity in the absence of any drug treatment. Caspase-3 activity was significantly inhibited when caspase-9DN was co-overexpressed, while the activity of the Apaf1 biosensor was significantly increased. Our results show that the Apaf-1 biosensor does not detect etoposide-induced apoptosis.

## 1. Introduction

Apoptosis is the process of programmed cell death in multicellular organisms and is vital for the development and regulation of homeostasis. Inappropriate apoptosis is associated with a wide range of diseases, from neurodegenerative diseases to cancer [1,2]. Two main/well-studied pathways through which apoptosis occurs are the extrinsic and intrinsic pathways. The extrinsic pathway is associated with the engagement of cell surface death receptors by their specific ligands, and the intrinsic pathway is associated with the release of cytochrome *c* from mitochondria in response to cell stress or damage [3,4]. In the mitochondrial-initiated pathway, cytochrome *c*, apoptotic protease-activating factor 1 (Apaf-1), and caspase-9 are the main components. Following a non-receptor-mediated stimulus produced inside the cell, cytochrome *c* is released through the formation of some pores in the outer mitochondrial membrane into the cytoplasm. There, it activates the adaptor protein Apaf-1, which then homo-oligomerizes to form a heptameric complex that recruits and activates procaspase-9. The interaction between Apaf-1 and caspase-9 is a homotypic one through their caspase recruitment domain (CARD). Active caspase-9 in turn activates caspase-3 and -7. These effector caspases can cleave a broad range of cellular targets and ultimately cause cell death [5,6]. Because of the importance of apoptosis in the developmental process and in human diseases, the mechanisms involved in this process and the development of assays to identify drug-like molecules that might be therapeutically useful have drawn a lot of attention in the field. To date, several assays suitable for high-throughput screening have been developed and applied for the detection of apoptosis. Each one relies on a specific feature of apoptosis pathway (whether intrinsic or extrinsic). However, until the development of the Apaf-1 split luciferase complementary assay, none could be used to monitor apoptosome formation within the cell death signaling pathway [7,8,9].

In this novel split luciferase reporter, Nluc/Apaf-1 and Cluc/Apaf-1, the N-terminal and C-terminal fragments of luciferase, are genetically fused to the N-terminal site of Apaf-1 [10,11]. Here, we extended these observations and investigated the ability of our split luciferase biosensor to detect apoptosome formation induced by other drugs besides doxorubicin. Results showed that etoposide, an effective apoptosis inducer [12], did not induce luciferase activity. However, we observed that overexpression of the Apaf-1 biosensor induced both luciferase complementation and caspase-3-like activity. To test whether the Apaf-1-induced cell death was apoptosome-dependent, we overexpressed a dominant negative form of caspase-9 (C287A). Caspase-9DN overexpressed with Apaf-1 split luciferase constructs enhanced luciferase activity while blocking caspase-3-like activity. These data suggest that the caspase-9DN makes the apoptosome more stable, possibly by preventing cell death [13,14,15].

## 2. Materials and Methods

### 2.1. Plasmids

pcDNA3-Casp9 C287A was a gift from Guy Salvesen (Addgene plasmid # 11819). Nluc/Apaf-1 and Cluc/Apaf-1 containing a flexible Gly–Ser linker in pcDNA3 were prepared according to a previous report [10].

### 2.2. Cell Culture, Transfection, Drug Treatment, and Cell Extract Preparation

First, 4.5 × 10^5^ HEK293T cells per well were seeded and cultured in 6-well plates containing 2 mL Dulbecco’s modified Eagle’s medium (DMEM, high glucose; Invitrogen) supplemented with 10% fetal bovine serum (Gibco) at 37 °C in an incubator with 5% CO_2_. Cells were transfected using Lipofectamine-3000 (Invitrogen) after reaching to a confluency of 70–90%. Twenty-four hours after transfection, cells were treated with different concentrations of etoposide. Drug-treated cells were incubated at 37 °C for different time periods, from 15 to 28 h. Cell extract was prepared using cell culture lysis reagent (CCLR, Promega). After removal of media, 100 µL of CCLR was added to each well, and the plates were incubated at 4 °C for 20 min. Then, cells were collected using a scraper, and the cell extract was driven after centrifuge for 2 min at 12,000 rpm. Vials containing the extracts were kept on ice to be used for split luciferase and caspase-3-like activity assays.

### 2.3. Cell Death and Cell Viability Assay

Cell viability was measured using alamarBlue assay. First, 2 × 10^4^ cells per well were seeded in 96-well plates in 200 µL of medium. Twenty-four hours after transfection, cells were treated with different concentrations of etoposide. Then, 24 and/or 48 h after treatment, 40 µL of 0.56 mM alamarBlue was added to each well, the plates were incubated at 37 °C for 5 h, and the fluorescence (λ_ex_ = 570 nm and λ_em_ = 600 nm) was measured. Cell death assay was measured by double staining of the treated cells with 1 mg/mL each of propidium iodide (PI) and Hoechst 33342 (H33342). Cell imaging and analysis was performed by Operetta High-Content Imaging System, and the percentage of cell death and apoptotic cells were calculated.

### 2.4. Caspase-3-Like Activity Assay

Caspase-3-like activity was measured using DEVD-AMC. First, 100 µL of the substrate (10 µM) was added to 15 µL of cell lysate. Fluorescent intensity was measured (λ_ex_ = 360 nm and λ_em_ = 460 nm) and normalized to the protein concentration of the lysate.

### 2.5. Luciferase Activity Assay

Here, 15 µL of cell lysate was mixed with 100 µL of luciferase assay reagent (Promega). Then, the produced light was detected immediately using a Victor plate reader at 25 °C over 15 min. Luminescent intensity was normalized by protein concentration.

### 2.6. Immunoblotting

Equal amounts of each sample (30 µg) was loaded on SDS-PAGE gel. After protein separation by gel electrophoresis, proteins were transferred to nitrocellulose membrane (100 v for 90 min). The membrane was blocked with 4% milk in TBS-T for 1 h at room temperature. Then, the membrane was incubated with primary antibodies, anti-Apaf-1 (AdipoGen, AG-20T-0134-c100), anti-caspase-9 (Cell Signaling Technology, 9502), and anti-actin (ProteinTech, 66009-1-Ig) overnight at 4 °C. The blot was washed 3–5 times for 10 min with TBS-T before incubation with secondary antibody (goat anti-rat; Li-cor-925–32219), goat anti-rabbit (Li-cor, 926–32211), and goat anti-mouse (Li-cor, 926–68020) in TBS-T. The membrane was then scanned using a LICOR system.

### 2.7. Statistical Analysis

A one-way ANOVA was carried out using GraphPad Prism version 8.3.0 to compare the treatment groups. Ns, *p* > 0.05; *, *p* ≤ 0.05; **, *p* ≤ 0.01; ***, *p* ≤ 0.001; ****, *p* ≤ 0.0001.

## 3. Results

The split luciferase complementary assay has previously been used to show the interaction of Apaf-1 molecules during apoptosome formation [9]. In addition, complementary luciferase activity has been detected in HEK293T cells transfected with split luciferase constructs of Apaf-1 (Nluc/Apaf-1 and Cluc/Apaf-1) and treated with doxorubicin, a known inducer of apoptosis [10]. Here, we tested the effect of etoposide as an alternative apoptosis inducer in HEK293T cells.

### 3.1. Etoposide HEK293T-Treated Cells Showed No Sign of Increase in Luciferase Activity

In order to obtain an effective concentration of etoposide for inducing cell death, a dose–response experiment was performed. An alamarBlue viability assay of untransfected HEK293T cells treated with etoposide showed concentration-dependent decreases in cell viability after 24 and 48 h (Figure 1a). After 48 h, 10 and 30 µM etoposide caused an ~75% decrease in viability as measured by this assay. However, the alamarBlue assay also reported decreases in cell number due to cell cycle arrest. As etoposide can cause S-phase slowing and an arrest in G2M, we tested whether 10 and 30 µM etoposide induced cell death assay using H33342 and PI double staining. Results showed ~25% cell death after 48 h incubation with 30 µM of the drug (Figure 1b).

We also investigated the cell death induced in cells transfected with green fluorescent protein (GFP), either untreated or treated with etoposide, to see the effect of transfection on cell death. As double staining with PI and H33342 gave a better measure of cell death than the alamarBlue assay, this assay was used (Figure 2a,b).

The percentage of dead cells was determined by scoring the percentage of PI-positive nuclei. As PI positivity occurs late in apoptosis, we also assessed the percentage of pyknotic nuclei using H33342 staining. This revealed that transfection significantly increased the percentage of PI-positive cells relative to untransfected cells, while the background level of pyknotic nuclei was ~30%. While etoposide treatment significantly increased the percentage of pyknotic nuclei in transfected cells, it did not significantly increase the percentage of PI-positive cells in transfected cells. These data suggest that there is a relatively high level of background apoptosis in cells transfected with the methods employed here.

Next, we tested the split luciferase activity in cells transfected with the Apaf-1 biosensor (Nluc/Cluc Apaf1). Cells were treated with 10 and 30 µM etoposide, and the luciferase activity was measured after 18 h. With drug treatment, no increase in split luciferase activity was observed compared to nontreated cells (Figure 3a). Indeed, etoposide-treated cells showed less luciferase activity in comparison with nontreated cells. We also treated cells with higher concentrations of etoposide and for both shorter and longer incubation times (10, 15, 24, and 28 h) but obtained similar results (data not shown). In parallel to measuring luciferase activity, we also assessed caspase-3-like activity in the same samples as a separate indicator of apoptosome activation. These measurements revealed that Nluc/Cluc Apaf1-transfected cells that were untreated with etoposide had about 10 times more caspase-3 activity than untreated cells that were transfected with pcDNA-luciferase (Figure 3b). These data suggest that overexpression of the Apaf-1 sensor expression is sufficient to induce caspase-3 activity in HEK293T cells, which is consistent with earlier reports that overexpression of Apaf-1 causes apoptosis [16].

### 3.2. Overexpression of Apaf-1 Sensor Causes the Activation of Caspase-3

To see if the concentration of constructs influences the function of the reporter assay, different amounts of Nluc/Apaf-1 and Cluc/Apaf-1 construct (250, 500, and 1000 ng of each construct) were transfected to HEK293T cells. Then, luciferase and caspase-3-like activities were measured. Increasing the amount of DNA increased both luciferase and caspase-3-like activity (Figure 4a,b). These data show that Apaf-1 overexpression can induce cell death, either because overexpression stresses the cell and activates the apoptotic machinery or because overexpression favors Apaf-1 oligomerization into a functional apoptosome. Based on this result, we hypothesized that transfection with a lower concentration of constructs might lower the background luciferase activity seen with Apaf-1 and allow the detection of apoptosome formation induced by etoposide. However, even when the expression of Apaf-1 was reduced, etoposide-induced luciferase activity could not be detected (data not shown).

### 3.3. Apoptosome Formation upon Overexpression of Apaf-1

Caspase-3 activation by Apaf-1 requires activation of caspase-9. We therefore tested whether expression of a dominant negative form of caspase-9 could affect luciferase activity and caspase-3 activity induced by Apaf-1 overexpression. Caspase-9DN was cotransfected with Nluc/Apaf-1 and Cluc/Apaf-1, and the expression level of all three proteins was assessed (Figure 5). Overexpression of caspase-9DN did not significantly increase the expression levels of Apaf-1. However, both caspase-3 and split luciferase activity showed a significant difference compared to the control (Nluc/Apaf-1 and Cluc/Apaf-1 as well as pcDNA empty vector). Overexpression of caspase-9DN caused a significant increase of ~4-fold in the split luciferase activity and a significant decrease of ~20-fold in the caspase-3-like activity (Figure 6a,b).

## 4. Discussion

In order to further optimize the split luciferase complementary assay of apoptosome formation, we tested etoposide as an apoptosis inducer because, like doxorubicin, it causes DNA damage through a mechanism that involves topoisomerase II. Like doxorubicin, etoposide is also reported to induce apoptosis through an Apaf-1-dependent mechanism. Despite the similarities with doxorubicin, no etoposide-induced luciferase complementary activity was seen in cells overexpressing Nluc/Apaf-1 and Cluc/Apaf-1. Although apoptosome-independent effects of etoposide have been reported [17,18], the more likely explanation is that, under the conditions tested here, overexpression of Apaf-1 masked etoposide-induced apoptosome formation.

Overexpression of Nluc/Apaf-1 and Cluc/Apaf-1 without any drug treatment caused a significant increase in caspase-3-like and luciferase complementation activities, possibly through induction of apoptosome formation, although an apoptosome-independent pathway cannot be excluded [19,20,21].

To test whether the observed luciferase and caspase-3 activities caused by overexpression of Apaf-1 constructs were apoptosome-dependent, we investigated the effect of a dominant negative form of caspase-9 (caspase-9DN). Caspase-9DN is recruited to the apoptosome but is catalytically inactive. This prevents the recruitment and autoactivation of wild-type caspase-9. Interestingly, caspase-9DN overexpression produced a large increase in luciferase activity and a decrease in caspase-3-like activity. The decrease in caspase-3 activity strongly suggests that the cell death induced by Apaf-1 overexpression relies on apoptosome formation. The increase in luciferase activity may be explained by caspase-9DN blocking cell death and allowing cells to survive with a high level of Apaf-1 expression that would otherwise cause cell death. It seems likely that the luciferase activity is due to Apaf-1 interaction in a functional apoptosome, but further experimentation is needed to demonstrate this.

## 5. Conclusions

Based on our findings, we can use overexpression of Apaf-1 sensor as a way of stress induction in HEK293T cells that might result in apoptosome formation and activation of caspase-3. It can therefore be used to examine the effect of inhibitors of apoptosome formation. In this way, high-throughput screening of different compounds affecting apoptosome formation would be much easier, more specific, and efficient. Furthermore, making a stable cell line expressing Nluc/Apaf-1 and Cluc/Apaf-1 together with caspase-9DN appears to be a simple approach to amplify the luciferase activity generated in cells expressing Nluc/Apaf-1 and Cluc/Apaf-1. This strategy could be used to improve the performance of high-throughput screens for inhibitors of Apaf-1–Apaf-1 interactions.

## Figures and Tables

**Figure 1 sensors-20-01782-f001:**
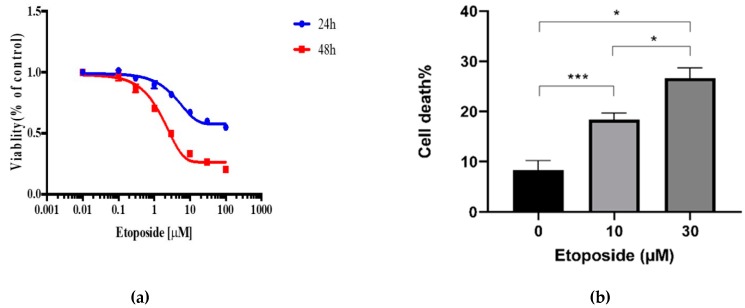
Viability and cell death assay to validate the effect of etoposide on HEK293T cells. (**a**) Cell viability assay was measured using alamarBlue reagent. Twenty-four hours after transfection, cells were treated with different concentrations of the drug. Then, 24 and/or 48 h after treatment, 40 µL of 0.56 mM alamarBlue was added to each well, and the plates were incubated at 37 °C for 5 h. Spectrophotometry of fluorescence was measured to indicate cell viability. (**b**) Cell death assay was measured using double staining of treated cells with 1 mg/mL each of propidium iodide (PI) and Hoechst 33342 (H33342). Cell imaging and analysis was performed 48 h after drug treatment by Operetta High-Content Imaging System, and the percentage of cell death was calculated. Results are the mean ± SEM of three separate experiments.

**Figure 2 sensors-20-01782-f002:**
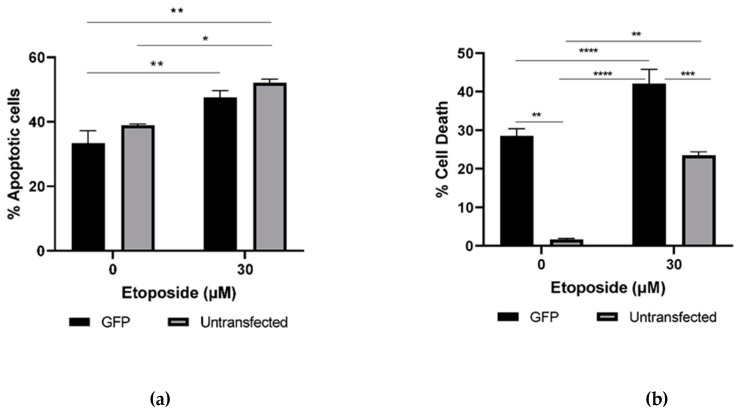
Effect of transfection on apoptosis and cell death in green fluorescent protein (GFP)-transfected cells treated with etoposide. The percentage of apoptotic cells and cell death were measured by double staining of treated cells (30 µM of etoposide for 48 h) with 1 mg/mL each of PI and H33342. Cell imaging and analysis was performed by Operetta High-Content Imaging System, and the percentage of apoptotic cells (**a**) and cell death (**b**) were calculated. Results are the mean ± SEM of three separate experiments.

**Figure 3 sensors-20-01782-f003:**
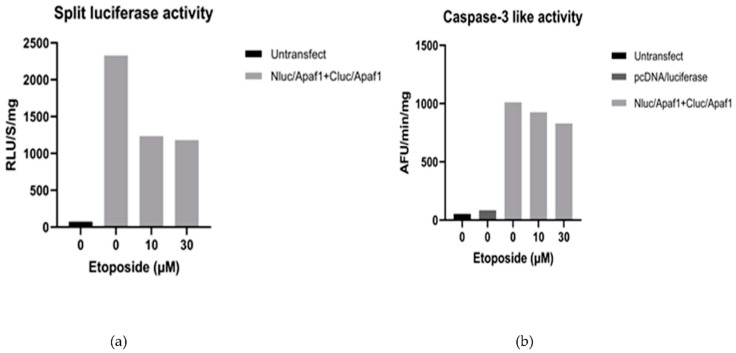
Split luciferase and caspase-3-like activity in apoptotic protease-activating factor 1 (Apaf-1) split luciferase-transfected HEK293T cells in the absence and presence of etoposide. HEK293T cells were transfected with 1 µg of each Apaf-1 construct. Twenty-four hours after transfection, cells were treated with 10 and 30 µM of etoposide. Eighteen hours after treatment, complementary luciferase activity (**a**) and caspase-3-like activity (**b**) were assayed. Graphs represent the result of one of the typical experiments.

**Figure 4 sensors-20-01782-f004:**
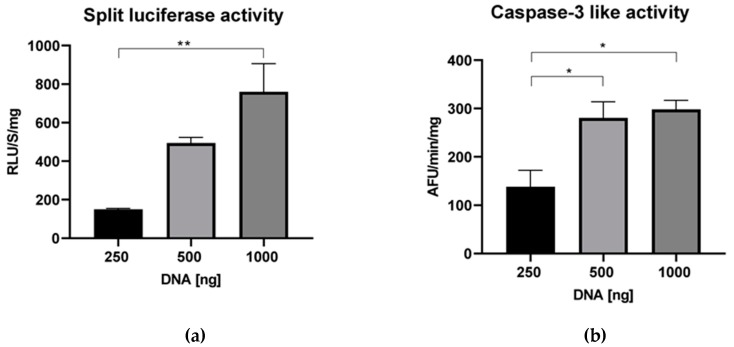
Concentration effect of Apaf-1 constructs on caspase-3-like and luciferase activity. HEK293T cells were transfected with 250, 500, and 1000 ng of each construct (Nluc/Apaf-1 and Cluc/Apaf-1). Forty-two hours after incubation at 37 °C, cell extracts were prepared, and both the split luciferase activity (**a**) and caspase-3-like activity (**b**) were assessed. Results are the mean ± SEM of three separate experiments.

**Figure 5 sensors-20-01782-f005:**
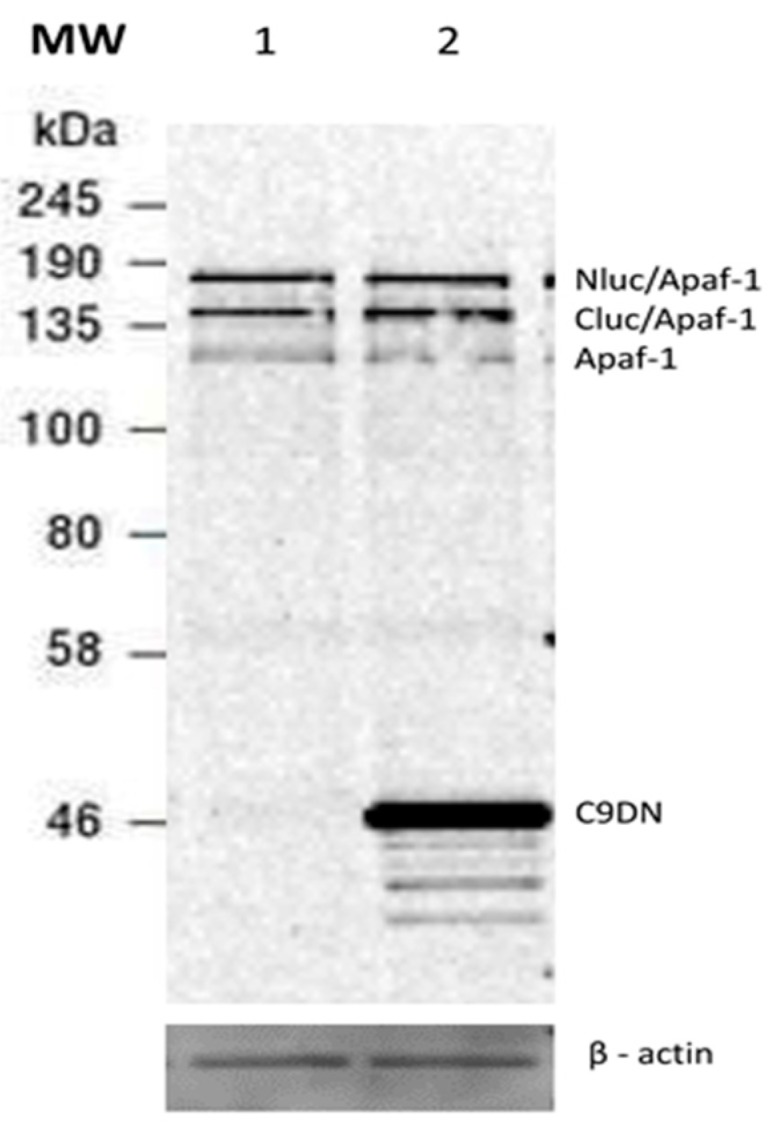
Immunoblot analysis of Apaf-1 constructs (Nluc/Apaf-1 and Cluc/Apaf-1) and caspase-9DN. Extracts of HEK293T cells transfected with 2500 ng of total plasmid (~830 ng of each DNA). Cell extract of each sample was prepared 42 h after transfection. The protein expression level for each construct was detected through immunoblot. Lane 1: cells transfected with Nluc/Apaf-1 + Cluc/Apaf-1 + pcDNA3; lane 2: cells transfected with Nluc/Apaf-1 + Cluc/Apaf-1 + C9DN. The result presents Western blot of one of three separate experiments.

**Figure 6 sensors-20-01782-f006:**
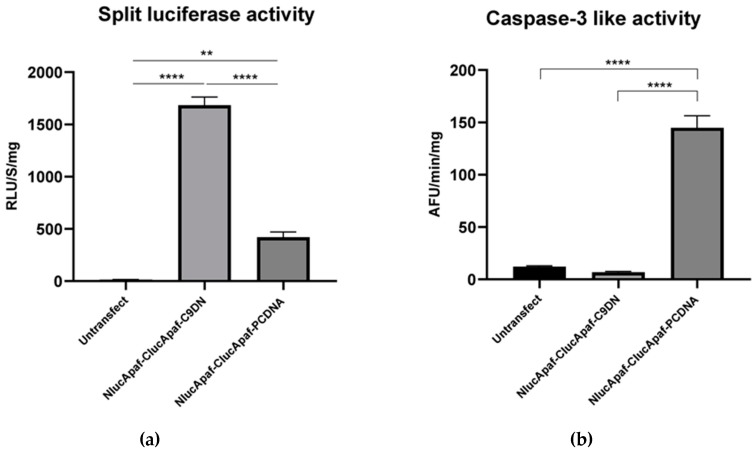
Reconstitution of luciferase activity and caspase-3-like activity in HEK293T cells overexpressing Apaf1 constructs and caspase-9DN. HEK293T cells were transiently transfected with 2500 ng of total plasmid (~830 ng of each DNA). Forty-two hours after transfection, cell extracts were prepared, and both luciferase (**a**) and caspase-3 (**b**) activity were assessed. Nluc/Apaf-1 + Cluc/Apaf-1+ C9DN compared with Nluc/Apaf-1 + Cluc/Apaf-1 + pcDNA. Results are the mean ± SEM of three separate experiments.

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
