# Peer review of "Optimization of Experimental Variables Influencing Apoptosome Biosensor in HEK293T Cells"

_sensors, 2020, doi:10.3390/s20061782_

Round 1
Reviewer 1 Report
This revised manuscript can be accepted.
Author Response
We thank the reviewer for positive response.
Reviewer 2 Report
After major revision, the quality of the manuscript has been improved. I suggest that this version can be accepted for publication.
Author Response
We thank for positive response.
Reviewer 3 Report
After reading the new version of the manuscript, I am pleased that the whole text have been edited and some figures have been modified to improve clarity. However, the authors didn't add any of the proposed experiments that I suggested including small modifications of figure like adding the molecular markers to the western blot. I still find the manuscript of low interest and confusing apart because of lack of many controls.
Despite these criticisms, I will support publication of the manuscript if the authors do those modifications:
1) the authors must express all their data as average+/- SEM and not SD.
2) the figure legend of Fig 5 and 6 must include the number of independent experiment used.
3) In page 7, last sentence of the paragraph: ''However, even when the expression of Apaf-1 was reduced, Etoposide induced luciferase activity could not be detected." The authors MUST present the data in the paper.
Author Response
We sincerely thank you for your precise and very helpful comments which we have tried our best to apply in present work and hopefully in future. All changes in the new manuscript are highlighted and Means of data are shown by ±SEMs in the current revised version. Figures are replaced with the revised figures .
Here you can find our responses to your last revision:
1.However, the authors did not add any of the proposed experiments that I suggested including small modifications of figure like adding the molecular markers to the western blot.
answer:We will try to expand our work by experimenting more controls including new mutations of Apaf-1 and new constructs of split luciferase (like Nluc/C9DN and Cluc/C9DN) in a future manuscript. The map of the MW marker has been added to Fig.5.
2.the authors must express all their data as average+/- SEM and not SD.
answer:It is done and now all graphs present data as average+/- SEM and not SD.
3.the figure legend of Fig 5 and 6 must include the number of independent experiments used.
answer: Thanks a lot for bringing this to our attention. It is done.
4.In page 7, last sentence of the paragraph: ''However, even when the expression of Apaf-1 was reduced, Etoposide induced luciferase activity could not be detected." The authors MUST present the data in the paper.
answer:The data for this result was not shown (and the phrase ‘data not shown’ is added).
Since we did not get any difference in Etoposide treated cells in lower amounts of plasmid concentration (250 and 500 ng of each construct) and no improvement was observed in luciferase activity compared to previous experiments, we gave up this experiment and moved to over-expression strategy independent of treatment with drug. As it is mentioned in discussion as you truly commented in previous revision, “Although apoptosome independent effects of Etoposide have been reported [17, 18], the more likely explanation is that under the conditions tested here overexpression of Apaf-1 masked Etoposide-induced apoptosome formation”. In addition, the reason that we did not test the lower amounts of Apaf-1 constructs (<250 ng) is that split luciferase activity signal in this concentration was so weak which is not suitable for our aim of using the reporter as a tool for screening compounds. So, we believe that because of these limitations in drug induced apoptosis, over-expression strategy of Apaf-1 biosensor would be a simple, specific and efficient way of apoptosome formation measurement which is currently being used in our lab instead of drug treatment.
However, for your consideration we have attached the raw data of the experiment and the graph presenting it as a file only for reviewer.
Round 2
Reviewer 3 Report
The authors fulfilled the recommendations and therefore I'm supporting this new version of the manuscript.
Round 1
Reviewer 1 Report
This manuscript investigated the apoptosis based on Apaf-1 molecules interactions with their CRAD domains, and concluded this may a useful tool for assay the chemical effect on aopptosome involved proteins. Results are interesting, but the novelty of this manuscript is not clear, some more experiments need to be carried out to confirm the conclusion. In additional, the authors should provide stained cell image in figures.
Author Response
We thank the reviewer for the comments. In fact, regarding the novelty of work as it has been clearly indicated in the revised version ( please see page 8 line 267-270) in the current work, apoptosome formation brought about independent of any chemical inducer (like Doxorubicin or Etoposide) and merely depending on overexpression of fused Apaf-1 constructs. Regarding the reviewer requests on requirement of more experiments to support the conclusion, we have to say that a set of Apaf-1 mutants are being made and under investigation to confirm the current conclusion and will be reported in a separate communication. Upon reviewer request for a stained cell image of Hek293 t cells upon exposure to Etoposide after 48 hours, a snapshot of one spot in one well has been shown here. Because of the low quality of HEK293T cell imaging (which are weakly stuck on the plate and do not give high quality images by our apparatus) they have not shown in the manuscript. However, if you feel they are required we can add them.
Reviewer 2 Report
In this article, Oladzad et al. report that over expression of Apaf-1 can be used as a kind of stress induction in HEK293T cells that might result in apoptosome formation and activation of caspase-3. Apaf-1 also could be used to evaluate the effect of inhibitors of apoptosome formation. What’s more, making a stable cell line expressing Nluc/Apaf-1, Cluc/Apaf-1 and Caspase-9DN might be a useful stable biological tool which allows to assay the effect of chemical compounds on apoptosome involved proteins. The work of this paper is clear and logical. However, I have decided against considering the paper for publication because of the following problems:
The method of this paper is not innovative enough. Most of the work is done by combining other people’s methods. Authors need to highlight their innovative contributions. Another obvious problem with this paper is the lack of sufficient experimentation to demonstrate the validity and applicability of the proposed method. Too few experiments make the conclusion of this paper lack persuasive. It’s noted that the manuscript needs careful editing by someone with expertise in technical English editing paying particular attention to English grammar, spelling, and sentence structure so that the goals and results of the study are clear to the reader. There are some grammar errors and typos in the manuscript. For example, “ml” should be “mL”, “µl” should be “µL”, please check. Besides, please make sure the picture format is consistent and meet the magazine publishing requirements. Some related latest reports are suggested to be cited to introduce the background.
Author Response
We thank the reviewer for the comments, please find below our responses and explanations for each issue:
1-The method of this paper is not innovative enough. Most of the work is done by combining other people’s methods. Authors need to highlight their innovative contributions. Another obvious problem with this paper is the lack of sufficient experimentation to demonstrate the validity and applicability of the proposed method. Too few experiments make the conclusion of this paper lack persuasive.
Our response: 1-We agree with reviewer on the lack of enough novel methods on design of current study. In fact, we have used a combination of different approaches in expansion of our previous works to optimize a biosensor for apoptosome formation in order to make it feasible for future applications. Moreover, as we have also asked by first reviewer, we have updated the novelty of the current investigation in the revised version. (please see page 8 line 267-270). Also, a set of Apaf-1 mutants are being made and under investigation to confirm the current conclusion and will be reported in a separate communication.
2- It’s noted that the manuscript needs careful editing by someone with expertise in technical English editing paying particular attention to English grammar, spelling, and sentence structure so that the goals and results of the study are clear to the reader. There are some grammar errors and typos in the manuscript. For example, “ml” should be “mL”, “μl” should be “μL”, please check.
Our response: 2- We have checked your points regarding the grammar errors and corrected them.
3- Besides, please make sure the picture format is consistent and meet the magazine publishing requirements. Some related latest reports are suggested to be cited to introduce the background.
Our responses: 3- The format of the pictures has been changed to a higher quality one considering meeting all the requirements of the journal.
Reviewer 3 Report
The manuscript entitled ''Optimization of experimental variables influencing apoptosome biosensor in HEK293T cells'' describes that an already published luminescence-based biosensor can trigger (without the need of apoptotic drugs) apoptosis or some aspect of it when over-expressed in cells. The Apaf-1 sensor seems a nice alternative to report early stage of apoptosis especially if use in an HTS setup. Unfortunately, I think that the presented manuscript is not suitable for publication in this present form.
The manuscript may have more impact if it was more clearly written and experiments presented/described in a better way and supported by appropriate statistical analysis. Moreover, this study is in complete contradiction with the original study describing the Apaf-1 sensor (ref #10). In the 2012 study, the authors saw a clear dose-dependent effect of doxorubicin on both luciferase and caspase-3 activities. In the actual manuscript, the authors stipulate that the lack of Etoposide effect on sensor activity may be explained by the fact that the drug action is not apoptosome dependent (last sentence of the first paragraph of the Discussion). An alternative explanation is that the authors simply over-expressed the sensor to levels that override the drug's effect. To be more convincing, the authors should repeat the experiment in Figure 4 in absence or presence of Etoposide or Doxorubicin. Also, the authors did not demonstrate convincingly that Apaf-1 over-expression has a significant impact on cell death.
Minor points:
Regarding data from Figure 3, the authors must show average of measurements and not only showing a typical data set. Also, the authors must show that both the control luciferase and the sensor are expressed at similar levels in order to compare them; for example, the authors can do this by western blot.
Regarding the legend of figure 4, (a) and (b) are in reverse position.
Figure 5 lacks molecular weight markers. Also, why the western blot does not show endogenous levels of cascase-9? The presence of this figure is not clear either.
The aim of Figure 6 is not clear; why using a DN mutant of cascase-9 and not simply measure the caspase activity?
Author Response
1-The manuscript may have more impact if it was more clearly written and experiments presented/described in a better way and supported by appropriate statistical analysis.
Our response: 1-The statistical analysis of all data is added to the revised version. A one-way ANOVA was carried out using GraphPad Prism version 8.3.0 to compare the treatment groups. Also, some revisions in writing especially in discussion and conclusion have been made.
2-Moreover, this study is in complete contradiction with the original study describing the Apaf-1 sensor (ref #10). In the 2012 study, the authors saw a clear dose-dependent effect of doxorubicin on both luciferase and caspase-3 activities. 3-In the actual manuscript, the authors stipulate that the lack of Etoposide effect on sensor activity may be explained by the fact that the drug action is not apoptosome dependent (last sentence of the first paragraph of the Discussion). An alternative explanation is that the authors simply over-expressed the sensor to levels that override the drug's effect.To be more convincing, the authors should repeat the experiment in Figure 4 in absence or presence of Etoposide or Doxorubicin.
Our response: 2-We agree with the reviewer on having an apparent contradiction with a former publication from application of drug point of view. In fact, in all previous transfections we saw a significant luciferase complementary activity in the absence of any cell inducer drugs. In order to characterize the essence of this luminescence activity we have come to this conclusion to see the effect of only overexpression of Apaf-1 constructs in the presence and absence of any cell death inducers. 3-However, despite significant effect of doxorubicin in induction of split luciferase complementary activity presumably through apoptosme formation at lower plasmid concentration, no significant activity was observed for Etoposide. This point has also been added to the revised version. (please see page 5, line 189-191 and page 7, line 236-240), that is why we suggested the probability of an apoptosome independent mechanism for Etoposide in HEK293T cells which requires more experiments to be confirmed.
4-Also, the authors did not demonstrate convincingly that Apaf-1 over-expression has a significant impact on cell death.
Our response: 4- Considering the importance of some more experiments in future which investigate the effect of Apaf-1 over-expression on cell death , here we more concentrate on apoptosome formation rather than cell death by measuring split luciferase and caspase3 activity (Fig 4a,b) and then we added Caspase9DN to investigate the involvement of Caspase 9 in apoptosome formation.
2
Minor points: 1-Regarding data from Figure 3, the authors must show average of measurements and not only showing a typical data set. Also, the authors must show that both the control luciferase and the sensor are expressed at similar levels in order to compare them; for example, the authors can do this by western blot.
Our response: 1-We did not have repeats with identical condition for the mentioned experiments in order to obtain an average of those experiments. However, a similar pattern was obtained under all repeated experimental conditions. Also this experiment which showed us a repetitive pattern of increased Caspase3 activity upon expression of sensor compared with control luciferase was just a starting point toward the hypothesis that the Apaf-1 sensor without the need of apoptotic drugs can trigger apoptosome formation, and the following experiments with data presented in Fig4-6 confirm the hypothesis.
2- Regarding the legend of figure 4, (a) and (b) are in reverse position.
Our response: 2- We thank the reviewer for bringing this point to our attention. The legend to figure 4 are corrected now.
3- Figure 5 lacks molecular weight markers. Also, why the western blot does not show endogenous levels of cascase-9? The presence of this figure is not clear either.
Our response: 3-Since expression of C9DN compared to endogenous one is significantly higher, so we took the photo with lower exposure to have a more qualified picture, therefore the band is very blurry. The original pic is attached here. Lane 3 and 4 from left which are related to the experiment have been cropped. As you see the endogenous C9 band is very faint but detectable and since our purpose was verification of C9DN expression, we used lower exposure.
4- The aim of Figure 6 is not clear; why using a DN mutant of cascase-9 and not simply measure the caspase activity? Our response: 4- We added Caspase9DN to investigate the involvement of Caspase 9 in apoptosome formation. The
reason that we used Cas9DN was that while it is recruited in apoptosome platform, prevents caspase3 activity and cell death (so, dominant negative effect of C9DN indicates its recruitment by apoptosome which confirms formation of apoptosome through expression of constrcts) , also leading to a more stable apoptosome complex which is a great option for optimization of biosensor . As we see a significant increase of split luciferase activity and decrease of Caspase3 activity gives us the opportunity of investigating apoptosome formation while having a stronger luciferase signal.